# The Identification Potential of Atherosclerotic Calcifications in the Context of Forensic Anthropology

**DOI:** 10.3390/biology13020066

**Published:** 2024-01-23

**Authors:** Sara Monteiro, Francisco Curate, Susana Garcia, Eugénia Cunha

**Affiliations:** 1University of Coimbra, Centre for Functional Ecology, Department of Life Sciences, Calçada Martim de Freitas, 3000-456 Coimbra, Portugal; sara.f.monteiro@inmlcf.mj.pt; 2National Institute of Legal Medicine and Forensic Sciences, 1150-334 Lisbon, Portugal; 3University of Coimbra, Research Centre for Anthropology and Health (CIAS), Department of Life Sciences, Faculty of Sciences and Technology, Calçada Martim de Freitas, 3000-456 Coimbra, Portugal; fcurate@uc.pt; 4Centro de Administração e Políticas Públicas, Instituto Superior de Ciências Sociais e Políticas, Museu Nacional de História Natural e da Ciência, Universidade de Lisboa, Rua Almerindo Lessa, 1300-663 Lisbon, Portugal; msgarcia@iscsp.ulisboa.pt

**Keywords:** forensic anthropology, identification, cardiovascular pathology, atherosclerosis

## Abstract

**Simple Summary:**

This study focuses on 71 human skeletal remains from the Luís Lopes Identified Collection, exploring atherosclerosis—and particularly calcified atherosclerotic plaques—in the context of forensic anthropology and the process of identification. Examining calcified atherosclerotic plaques can be valuable for confirming or disproving an identification, but this approach is contingent on the availability of pre-mortem imaging exams. It should always complement other identification methods for optimal accuracy.

**Abstract:**

Atherosclerosis is an inflammatory disease that, in its more developed stages, can lead to the calcification of fatty plaques on the walls of arteries, resulting in the appearance of new bone elements. It is a condition that has been studied and documented little in the context of paleopathology, especially in the framework of forensic anthropology. This article analyzed the skeletal remains of 71 individuals (35 females and 36 males) from the Luís Lopes Identified Collection of the National Museum of Natural History and Science in Lisbon, 31 of whom had an autopsy report. An attempt was made to ascertain whether these bone elements resulting from atherosclerotic calcification would resist cadaveric decomposition and whether they would be recoverable several years after burial, and a survey was carried out of their distribution according to sex and age, as well as their association with other pathologies, such as osteoporosis and cardiac and renal pathologies. An imaging analysis of an atherosclerotic plaque was also carried out to complement the macroscopic analysis and present other methods of identifying plaques. It was concluded that each atherosclerotic calcification has a unique profile, which can be useful for identification, especially in cases where the individual shows a severe condition. In terms of identification potential, the analysis of calcified atherosclerotic plaques can be useful, as they can corroborate or reject an identification. However, it always requires the existence of ante-mortem imaging exams and must always be used in addition to other identification methods. Regardless of the identification, these plaques are bone elements resulting from a pathology and should, therefore, be known and recognized by the scientific community.

## 1. Introduction

Forensic anthropology investigates human remains in different states of decomposition, including those saponified, skeletonized, mummified, burned, and/or fragmented, and living individuals. This article will focus on the necro identification process—i.e., the assessment of the specific identity of an anonymous individual—in forensic anthropology to evaluate a possible supplementary identification feature: vascular calcifications, specifically atherosclerotic calcifications. 

Atherosclerosis is a disease that affects the tunica intima of large and medium-caliber elastic arteries and muscular arteries. It is considered an atheromatous disease due to the presence of irregular intimal plaques (the atheromas) that endure in the lumen of the arteries (coronary, carotid, and cerebral), the aorta and its branches, and the large arteries of the limbs, and that are formed by lipids, muscle cells, and a connective tissue matrix that may contain calcium deposits. Atheromatous plaques can spread to all medium and large arteries but generally begin where arteries branch, bifurcate, or bend, and where blood flow is more irregular [1]. Atherosclerosis and the calcification of its plaques lead to a progressive narrowing of the arterial caliber (stenosis) and tend to affect the elastic characteristics of the vessels. Doherty et al. [2] suggest that vessel calcification does not occur through passive calcium precipitation but due to an active and regulated process that is similar to osteogenesis. This hypothesis arises from the fact that a calcified artery does not only contain calcium but also other components that can be found in bones, such as hematopoietic tissue and hydroxyapatite crystals.

In forensic anthropology, the identification process begins with the assessment of the biological profile, i.e., the estimation of population affinities, biological sex, age at death, and stature, as these four parameters are the most generic in terms of identity. The methods to estimate these parameters are procedurally heteroclite and are based on different regions of the human skeleton, thus safeguarding diverse recovery circumstances. Establishing these four elements allows for many individuals to be excluded from the outset, yet the list of possible identity suspects can nonetheless remain extensive. The second step is to investigate individualizing characteristics, i.e., anatomical, pathological, or traumatic features that show the potential to differentiate a skeleton from all the others. These variations are numerous and diverse, and the rarer the feature, the higher its potential for identification. It is important to note that the effectiveness of the identification procedure often relies on the condition of the human skeletal remains and the availability of relevant records or reference materials for comparison. In many cases, a multidisciplinary approach involving different methods is employed to enhance the accuracy of identification.

Several pathological conditions cause lesions in the skeleton. These pathologies are wide-ranging, even if the skeleton’s response to illness is constrained, and include degenerative, infectious, metabolic, or circulatory diseases, among others. Determining how the effects of any pathology impact an individual’s daily life is sometimes more important than identifying it [3]. Furthermore, when surveying for signs of pathology in the skeleton, as in the search for other elements, the fact that no trace is observed does not mean that no pathological condition occurred and affected the individual. 

In addition to the difficulty of diagnosing a pathology that affects the skeleton, it was only very recently that atherosclerosis was included in the range of diseases that can be detected by analyzing decomposed or skeletonized human remains. Calcifications are not exclusive to atherosclerosis, and different diseases can lead to the formation of various types of calcifications [4,5]. On the other hand, atherosclerotic calcifications are a vestige of atherosclerotic disease and, therefore, fall into the group of pathological variants. Even though they are a direct result of cardiovascular disease (circulatory disease), they are usually “ignored”, either because the calcified plaques are not in situ or mainly because they have not been adequately studied [6]. The likelihood of identifying an individual from the analysis of skeletal lesions increases considerably when there are medical records and, more specifically, when imaging assessments are performed during the lifetime (preferably not long before death) since these can be replicated and superimposed, allowing the confirmation or rejection of a presumed identity [7,8]. Atherosclerosis confers an identical potential for identification, but only as a complementary method following standardized individualization procedures.

The calcified atherosclerotic plaques (CAPs), in addition to not being present in all individuals, vary in size, shape, and location and are therefore different from person to person. Given these characteristics, their individualizing potential is high. This study will, therefore, focus on CAPs, seeking to shed light on atherosclerosis in general, but more specifically on their identification, preservation, population distribution, and whether they can be considered an identifying characteristic in forensic anthropology.

## 2. Materials and Methods

### 2.1. This Study-Sample 

This study-sample stems from the Luís Lopes Collection, also known as the Lisbon Identified Skeletal Collection, curated at the National Museum of Natural History and Science (MUHNAC) of the University of Lisbon. The collection results from an agreement between the municipality and the museum, in which MUHNAC obtained authorization at specific times to collect some skeletons destined for incineration. It is an identified skeletal collection that includes 1673 identified skeletons, although there is no complete information for all of them. Most of the individuals were Portuguese nationals who were born between 1805 and 1972 and died between 1880 and 1975 in the city of Lisbon. For approximately 750 individuals, there is a set of information, such as biological sex, place of birth, date of death, marital status, occupation, and the cemetery from which they were exhumed [9]. The existence of biographical data, made available to researchers only in exceptional circumstances of research, allows them to search for additional biographical or medical information, such as, in the case of this investigation project, the autopsy reports, whether in clinical or medico-legal contexts.

Of the 1673 identified skeletons in the collection, 228 were autopsied, and given that access to autopsy reports was necessary for the development of this topic, the only inclusion criterion was having been autopsied. Two exclusion criteria were also established to make the sample more appropriate to the objectives, and the following were excluded: (1) individuals under the age of 30 (since the pathology under study is more frequent in older individuals [10]), and (2) the cases in which autopsies were not performed at the South Delegation of the National Institute of Legal Medicine and Forensic Sciences (NILMFS) or at São José Hospital (Lisbon, Portugal). The legal and ethical consents pertaining to this research apply only to those two institutions.

After applying both inclusion and exclusion criteria, the final study-sample comprised 71 skeletons (35 females and 36 males). Moreover, a sub-sample consisting of the 31 individuals for whom it was possible to access the autopsy reports (15 females and 16 males) was created. In this sub-sample, the data obtained from skeletal analysis and the data obtained from the autopsy reports were compared, assessing their similarities and discrepancies. 

Additional calcifications were used to perform an imaging assessment: a biliary calcification, a renal calcification, and an atherosclerotic calcification. These calculations are from three individuals who were accessed in a professional context and whose identification data were entirely anonymized. Before the calcifications were collected, the National Registry of Non-Donors (RENNDA) was consulted to ensure that none of the individuals in question opposed donation. The reason for their inclusion is that, during their autopsies, vascular and other types of calcifications were detected in an advanced stage of development, thus showing a potential for comparison and illustration of the subject.

### 2.2. Methodological Approaches

The methods used were essentially based on macroscopic analysis. First, an anthropological study was made of each of the individuals in the sample, starting with an analysis of the bone parts of each skeleton. The emphasis was placed on the nutritional foramina since blood vessels pass through them and are sometimes subjected to calcification [11,12]. Calcified atherosclerotic plaques were recorded as absent or present. 

According to bone representativeness and preservation, a classification was given to each of the skeletons, with four possible levels defined, similar to Biehler-Gomez et al. [6]: Optimal: when both bone preservation and representativeness are above 90%; good: when both factors are above 60%; average: when they are between 40% and 60%; poor: when at least one of the factors is only around 40% or less.

Signs of specific pathologies that could be related to the presence of atherosclerosis, specifically osteoporosis, heart disease, and kidney disease, were surveyed [13,14,15,16,17,18,19]. Osteoporosis was not clinically diagnosed but only identified macroscopically and qualitatively in the entire sample. The classification criteria for osteoporosis were: identification of osteoporotic fractures of the vertebrae, distal radius, proximal humerus, and femoral neck; and assessment of weight and bone fragility [20]. Heart and kidney disease were retrieved from the autopsy reports. The presence of other calcifications was also studied, such as those in the thyroid cartilage, cricoid, and ribs, to ascertain if there was any relationship with their presence or absence.

Next, the small bone fragments that remained on this study table were observed in detail, and it was at this stage that most of the CAPs were detected. Any elements that could be identified as plaques were compared with the images provided by Biehler-Gomez et al. [6,21] in their scientific articles focused on this study topic. If identified as CAPs, they were also classified according to their tubular or concave shape, considering that concave CAPs can result from the fragmentation of tubular CAPs. Finally, they were photographed using a Canon EOS 6D Mark II (Canon Inc., Ōta, Tokyo, Japan) and a ProScope HR2 (ProScope Digital, Wilsonville, OR, USA), measured using a digital caliper, and the measurements were entered on the record sheet.

To distinguish CAPs from other types of calcifications, some of the most common elements with which atherosclerotic plaques could be confused were demarcated, and some criteria for differential diagnosis were defined for these elements. Several clinical conditions lead to calcification of the pulmonary pleurae, including tuberculosis and other infectious diseases [22], cancer, carcinoma, and pulmonary hamartoma, among others [23]. Similarly, dural calcifications, teratomas, and costal cartilage calcifications could also interfere with the identification and classification of plaques as atherosclerotic [24]. 

The calcifications resulting from these pathologies have different configurations, but in general, the primary distinction made between CAPs is their thickness and surface irregularity. A calcified atherosclerotic plaque has slightly irregular margins, as does its surface, but even in a macroscopic analysis, a stratified deposition of layers of bone is visible. On the other hand, pleural calcifications in particular, although macroscopically they are also a superimposition of layers of bone, are considerably more irregular, resulting in thicker calcifications than atherosclerotic ones [25]. Concerning rib cartilage calcifications, these are also more irregular than CAPs, both in terms of shape and surface. Of all the calcifications mentioned above, only dural calcifications do not overlap with CAPs in terms of thickness since dural calcifications are usually thin and smooth [26].

In the case of the sub-sample, which includes information from autopsy reports, once the skeletons had been fully studied, the respective reports were searched for at the INMLCFs southern office or at the São José Hospital. This involved filling out the second part of each individual’s registration form, which includes age, cause of death, references to CAPs in the various arteries, and remarks regarding additional pathologies, as well as other relevant information.

Finally, the three additional calcifications mentioned before (a biliary calcification, a renal calcification, and an atherosclerotic calcification) were selected for imaging analysis using computed tomography (CT), which was carried out at the Coimbra University Hospitals. Two images were obtained for each of the calcifications according to the image processing techniques applied. Two types of three-dimensional reconstruction were used for the biliary and renal calcifications: shaded surface display (SSD) and volume rendering technique (VRT). For the atherosclerotic calcification, two variants of VRT were used, given the greater level of detail [5].

After collecting the data from the skeletons and the respective autopsy, the data obtained were processed with SPSS (Statistical Package for the Social Sciences) (version 25.0). A Chi-Square test, which in one case involved applying the Monte Carlo correction (which was applied in one case when the conditions for a chi-squared test were not met), and Fisher’s exact test were enacted in order to compare the frequency distribution of a variable between two groups. The Phi correlation coefficient was used to assess the strength of the association between two variables. Finally, the Kappa statistic was used to evaluate the agreement between the observations and the autopsy reports [27,28,29,30,31,32,33].

## 3. Results

In the analysis of representativeness and bone preservation parameters, according to the proposed classification categories, zero (0%) individuals presented an optimal classification, 30 individuals (42.25%) had good bone preservation and representativeness, 31 individuals (43.66%) had average preservation and representativeness, and ten individuals (14.08%) had poor bone preservation and representativeness.

Of the 71 individuals in the sample, 43 CAPs were detected in 26 individuals (36.6% of the sample). Of all the plaques recovered, 33 were concave (Figure 1), and 10 were tubular (Figure 2). All but two of the plaques represent macrocalcifications, as they are larger than 2 mm.

The proportion of females (11) and males (15) with CAPs was very similar in both the sample (chi-square: 0.010, *p* = 0.920) and the sub-sample (Figure 3). 

As expected, atherosclerotic plaques were more prevalent in older individuals (Figure 4), especially after the sixth decade of life.

To establish a relationship between atherosclerosis and osteoporosis, although the figures obtained are not in themselves indicative of any relationship between the two pathologies, in statistical terms, this relationship was substantiated, given that Fisher’s exact test had a result of *p* = 0.003 and subsequently the Phi correlation coefficient obtained a value of 0.369, confirming a statistical relationship between the two variables [27,28,29,31,32]. The same happened with heart conditions, although the results only covered a sub-sample. Fisher’s exact test gave a result of *p* = 0.007, and the Phi correlation coefficient gave a result of Ф = 0.538, also confirming a relationship between the variables. Only kidney disease had no statistical result to support the theory of a possible relationship with atherosclerosis, as Fisher’s exact test had a *p*-value of 0.535, confirming that there was no statistically significant relationship between the variables. Regarding the presence of atherosclerosis simultaneously with other types of calcifications, a relationship was found with cartilage calcification, and of the 24 individuals with calcification of the various cartilages considered, 19 also had atherosclerotic calcifications. Statistically, a Chi-Square test was obtained with a *p*-value of 0.001, and the strength of the correlation was given by the Phi correlation coefficient, which resulted in Ф = 0.404, thus characterizing a solid relationship between the variables [27,30].

Since there was a sub-sample with skeletons and autopsy reports, the Kappa statistic was applied to check whether the observations made on the skeletons regarding the presence of PACs agreed with the information contained in the respective autopsy reports. The Kappa value obtained was 0.516, which means that there is substantial agreement between the researcher’s observations of CAPs on the skeleton and the medical experts’ observations of CAPs during autopsies [27,29].

## 4. Discussion

This study was limited by the fact that the collection of autopsy reports, most of them stored in a hospital, was interrupted by the pandemic, as well as the loss of some of those reports with water damage, which resulted in a limited number of cases with autopsy reports available (*n* = 31). Thus, of a total of 71 skeletons analyzed, only a sub-sample of 31 possessed autopsy reports. Additionally, the context in which the bones were recovered may be the reason why some smaller bones, for example, from the hands and feet, and the CAPs themselves, if existent, were not recovered since the exhumations were carried out by cemetery technicians and the clothes were lost, so there was no control over the representativeness of the bones [9]. This type of recovery always constrains the collection efforts; when bones are lost, information is also lost.

In our study, the atherosclerotic plaques recovered are tiny (less than 2 cm), nonetheless identifiable and distinct from other calcifications. The imaging assessment of the biliary and renal calcifications that were carried out in this study shows a distinct pattern from the atherosclerotic calcification.

As research into atherosclerotic plaques in forensic anthropology is a relatively new topic, it is not possible to properly compare the results that have been obtained in this study. To our knowledge, the only study available to this effect is that by Biehber-Gomez et al. [6] in a sample from the Milano Cemetery Skeletal Collection. In both studies, males and females seem to be similarly affected by CAPs. Likewise, the results of both studies suggest that these calcifications tend to occur in older individuals. Regarding the prevalence of plaques, the total of 43 CAPs in 26 individuals (out of a total of 71 individuals) corresponds to a percentage of around 40%, and, on average, each individual had less than two plaques. This prevalence is roughly half of that observed in the sample from the Milano Cemetery Skeletal Collection. A statistical correlation was found between atherosclerosis and osteoporosis, as well as between calcification of neck and rib cartilage and cases with atherosclerotic plaques. The association of atherosclerosis and osteoporosis highlights the simultaneous progress of the two pathological processes leading to tissue damage, resulting in elevated occurrences of both fatal and non-fatal coronary events, along with an increased risk of fractures [16,17,18]. Additionally, it should be noted that all the cases with plaques have been noted in the respective autopsy reports.

Notwithstanding, several factors prevent a proper comparison, such as the different conditions of access and context in which the various elements were analyzed and processed and the differences in sample size, as mentioned above [34]. 

We are aware that the small sample size prevents any major conclusions or comparisons, but we would nevertheless like to stress the importance of this study in order to draw attention to the importance of reporting atherosclerotic plaques during forensic examinations. There is no doubt that these plaques can reveal important medical information, which is crucial for individualizing factors.

This article provides further evidence that forensic anthropologists must strive to go beyond the skeleton and bones when investigating the biological profile of unidentified human remains, since other biological remnants, namely biological calcifications, can be utterly relevant for the identification process [35]. As expected, these calcifications are valuable extra-skeletal features under different circumstances and can serve multiple purposes, namely the corroboration of the biological profile. In fact, they are more common at advanced ages, and thus their presence points toward elderly people. Moreover, they can serve as useful identity factors. Artery calcifications are a direct consequence of vascular conditions, and as a result of their calcified nature, they can survive the decomposition process; thus, these biological elements are sometimes available for the anthropological exam. That said, despite atherosclerosis being a soft tissue disease, diagnosing it in skeletonized remains might be possible, with major consequences for the forensic identification process. 

Calcified atherosclerotic plaques are rare, and, at least to a degree, their uncommonness can be explained by identification shortcomings and inadequate recovery strategies. Some practical recommendations should be given in order to enhance the recovery of CAPs in skeletal remains from forensic contexts, as their potential to improve identification is indisputable. They usually present as small pieces; as such, meticulous observation, recording, and sieving of the skeletal remains must be accomplished on-site and also during forensic expertise. Also, the body bag should be carefully examined and, in most circumstances, thoroughly sieved. Special care must be reserved for the process of undressing the body since these biological remains may remain attached to the clothes. In addition, the cleaning procedure for these calcifications can be delicate and complex. In that respect, we recommend following the instructions provided by Biehler-Gomez et al. (2021) [35]. 

Interestingly, calcified blood vessels also provide evidence of a lengthy diachronic prevalence of atherosclerosis in human populations. To our knowledge, the oldest cases in which atherosclerotic plaques were detected were associated with skeletal remains recovered in Amara West, Sudan (1300–800 BC) [36]. Even though the present work focuses on the importance of CAPs for forensic practice, as they can add value to the expertise during the identification process, the results are undoubtedly appealing to the study of health and disease in past populations.

## 5. Conclusions

Identifying anonymous human skeletal remains in the context of forensic anthropology poses several methodological challenges due to various factors, and additional markers of identity can contribute to the overall forensic reconstruction of an individual’s life history, fostering the potential for individualization. 

As such, in terms of identification potential per se, it is hoped that calcifications can contribute to the individualization of unidentified skeletal remains, especially if the calcifications are recovered in situ and present with a considerable size. The data from this study suggests that atherosclerotic plaques alone cannot contribute to the identification of anonymous individuals, but that they can corroborate or challenge an attempted identification, particularly in older individuals, as well as provide some circumstantial information that can be useful for understanding the individual’s biographical context. Regardless of their capability to contribute to identification, it is important to recognize and classify CAPs, particularly their typical physical manifestations and the contexts in which they are usually recovered, as they are still elements that can provide relevant information about unidentified individuals.

## Figures and Tables

**Figure 1 biology-13-00066-f001:**
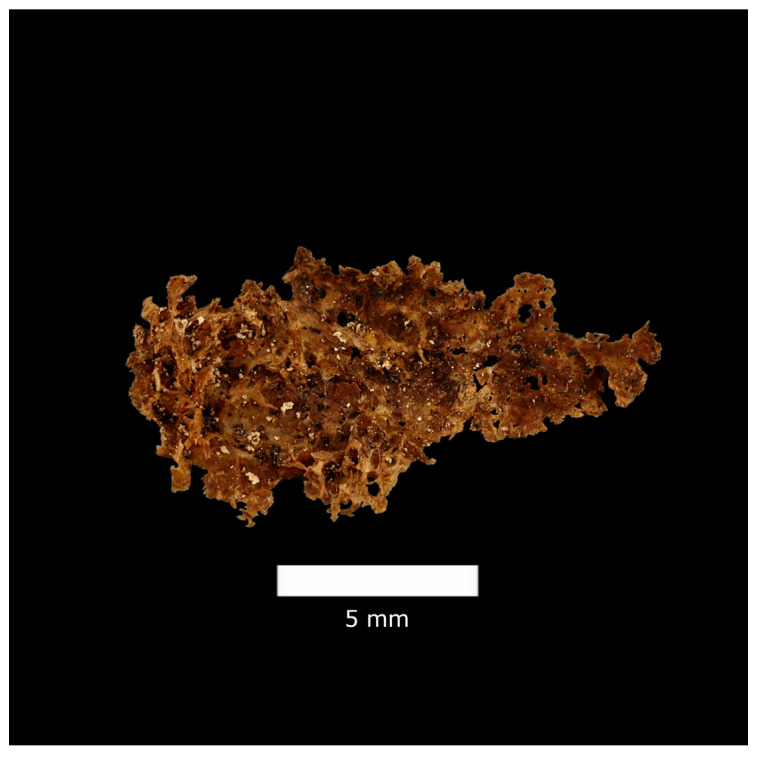
Concave CAP, MNHNC:MB61:001457 (female, 73 years old), Luís Lopes Collection, Universidade de Lisboa (image: R. Keller, Universidade de Lisboa).

**Figure 2 biology-13-00066-f002:**
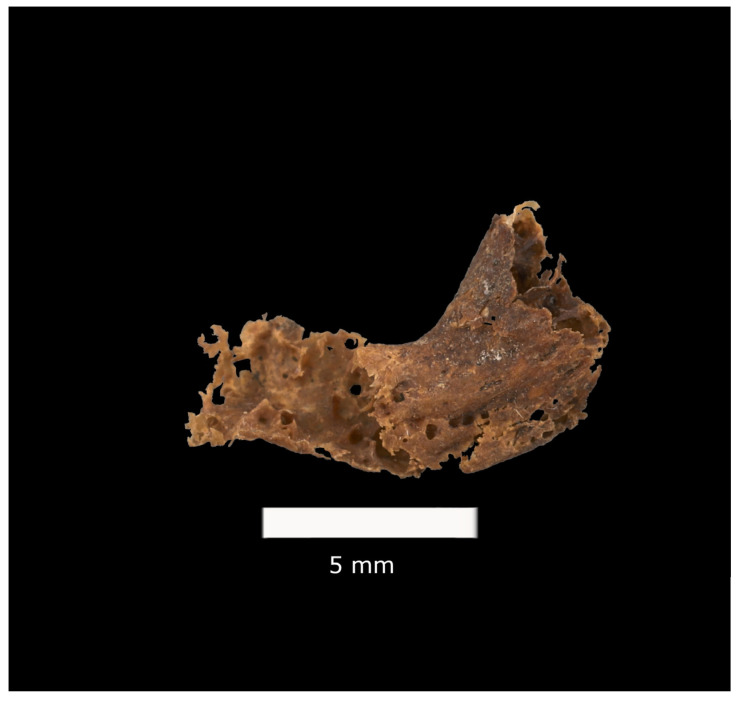
Tubular CAP, MNHNC:MB61:000589 (female, 64 years old), Luís Lopes Collection, Universidade de Lisboa (image: R. Keller, Universidade de Lisboa).

**Figure 3 biology-13-00066-f003:**
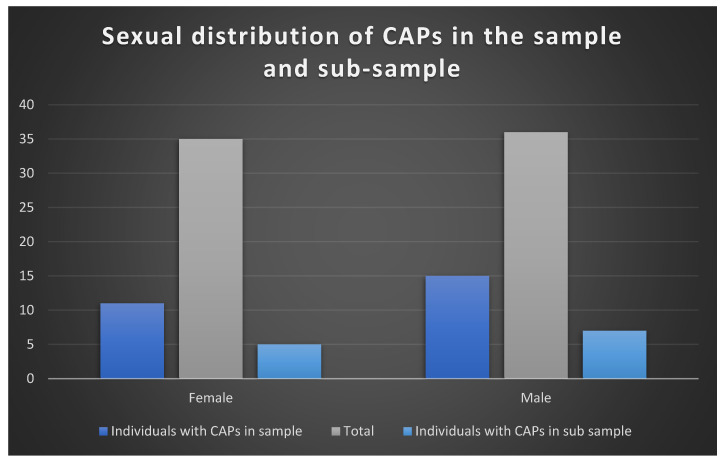
Sexual distribution of CAPs in the sample and the sub-sample.

**Figure 4 biology-13-00066-f004:**
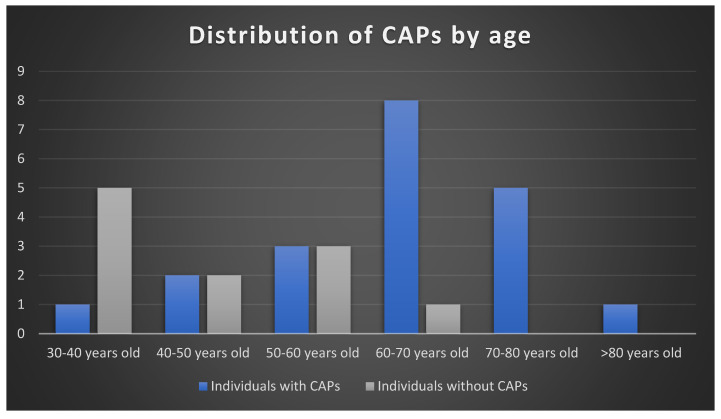
Distribution of CAPs by individuals’ age in the sample.

## Data Availability

The data presented in this study are available on request from the corresponding author. The data are not publicly available due to ethical reasons.

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
