# Peer review of "The Identification Potential of Atherosclerotic Calcifications in the Context of Forensic Anthropology"

_biology, 2024, doi:10.3390/biology13020066_

Round 1

Reviewer 1 Report

Comments and Suggestions for Authors

This is an interesting paper looking at the calcifications from atherosclerosis in order to determine if they are uniquely identifiable to a specific individual. The results suggest more research should be conducted in this area as this sample showed promise. I like this paper because when I worked in autopsy I wondered if this would be useful for identification and wanted to do a study on it. However, I didn’t have time and forgot about the idea. I am so glad to see someone do this! I only have a few comments, detailed below.

Page 4, last paragraph before Results: Please elaborate further on your statistics. What were your variables and how were they coded (e.g., present/absent)? What does it mean “which in one case involved applying the…”? I read that as all of those statistics were applied to one individual, which can’t be correct.

Page 5, line 209: do you mean “chi-square”?

Author Response

We are indebted to your considerate remarks and thoughtful suggestions. We have made the revisions accordingly.

Page 4, last paragraph before Results: Please elaborate further on your statistics. What were your variables and how were they coded (e.g., present/absent)? What does it mean “which in one case involved applying the…”? I read that as all of those statistics were applied to one individual, which can’t be correct.

We have added more information according to this remark and clarified the unclear sentence.

Page 5, line 209: do you mean “chi-square”?

Corrected.

Reviewer 2 Report

Comments and Suggestions for Authors

The article intitled “The Identification Potential of Atherosclerotic Calcifications in 2 the Context of Forensic Anthropology” is a very interesting study that brings to the fore elements that are little studied in forensic anthropology.

Here some suggestions and comments below:

Material:

p.3 l.120-121: the authors can remove the parentheses.

l.122-130: the addition of the 3 individuals is there to serve as a point of comparison for the elements that will be observed on the 71 individuals included in the study? Could you clarify this a little further?

Methods:

What is the purpose of these statistical tests? The authors can specify the purpose of these analyses.

Part of the method presented is not found in the results, for example representativeness. it would be a plus to have all the results.

Results:

The figures 1and 2 are a little blurred. The photos should be of higher quality for better visibility.

In the figure 3, the sub-sample is shown. To make it easier to understand, you can add a word to the text (l.208 p.5).

l.209 p5: “qui-square result“: I think some of the information is missing.

Discussion:

From the results onwards, there is no longer any mention of the 3 individuals added. Why is this? Perhaps the authors can provide more justification, as we wonder what the point of this addition is.

Even if comparisons are difficult to make due to lack of literature or a small sample size, the authors can still suggest trends in relation to Biehler-Gomez's study?

In the results section, the kappa with the sub-sample is discussed. no discussion is provided afterwards. what are the differences in observation of the CAPs?

All in all, a very interesting and necessary study for the discipline. However, there are certain elements that could be developed further to strengthen the work and exploit all the results.

Author Response

We are thankful for your discerning clarifications and comments.

p.3 l.120-121: the authors can remove the parentheses.

Removed as suggested.

l.122-130: the addition of the 3 individuals is there to serve as a point of comparison for the elements that will be observed on the 71 individuals included in the study? Could you clarify this a little further?

We thank you for this remark; as suggested, we have clarified this issue by saying that these individuals will serve as a comparison and illustration of the subject.

Methods:

What is the purpose of these statistical tests? The authors can specify the purpose of these analyses.

We have added more information as requested.

Part of the method presented is not found in the results, for example representativeness. It would be a plus to have all the results.

We do agree, thank you. We have thus provided the following statement:

"In the analysis of representativeness and bone preservation parameters, according to the proposed classification categories, zero (0%) individuals presented an optimal classification, 30 individuals (42.25%) good bone preservation and representativeness, 31 individuals (43.66%) average preservation and representativeness and ten individuals (14.08%) poor bone preservation and representativeness."

Results:

The figures 1and 2 are a little blurred. The photos should be of higher quality for better visibility.

New images, with better quality, were added.

In the figure 3, the sub-sample is shown. To make it easier to understand, you can add a word to the text (l.208 p.5).

Added.

l.209 p5: “qui-square result“: I think some of the information is missing.

The result of the test was added.

Discussion:

From the results onwards, there is no longer any mention of the 3 individuals added. Why is this? Perhaps the authors can provide more justification, as we wonder what the point of this addition is.

We regret that the inclusion of these three individuals was not explicit. They were included to provide a control/comparison.

Even if comparisons are difficult to make due to lack of literature or a small sample size, the authors can still suggest trends in relation to Biehler-Gomez's study?

As suggested, a paragraph was added to compare the main results with the Biehler-Gomez study briefly.

In the results section, the kappa with the sub-sample is discussed. no discussion is provided afterwards. what are the differences in observation of the CAPs?

We haven’t addressed Kappa's results since we have found and stated that the results showed good agreement, and further discussions seem unnecessary.

Reviewer 3 Report

Comments and Suggestions for Authors

First three paragraphs of the introduction are in fact long-winded presentation of the topic. Instead, author should rewrite it and compile 2-3 sentences in a single paragraph giving out the topic. This is probably 15th paper I"m reviewing that attempts to establish anthropology - in vane. The term "identification" should be defined better. 

How was the sex determinedin in ln 116?

 Some parts of the "introduction" belong to rhe "methods", and the methodology section should be sub-divided tnto "subheadings" (2??). 

I believe that lns 86-88 comprise whole paper and my oppinion on it. If anything - this should be the "concllusion". Discussion is weak and mistakej for the "conclusion". Discussion should contain  the results and outcomes of a study. An effective discussion informs readers what can be learned from your experiment and provides context for the results

"Calcified atherosclerotic plaques" are "CAP" in most cases, but somewhere (as in ln 89) it is given in full 

Ln 115 - "shortened sample"???

Figures 1 and 2 are of a low quality and blurred - too poor for a publication and outlook/ color scheme should be changed. In figs. 3 and 4.

Comments on the Quality of English Language

Style and wording should be improved!

Author Response

We are grateful for the judicious observations and amendments.

First three paragraphs of the introduction are in fact long-winded presentation of the topic. Instead, author should rewrite it and compile 2-3 sentences in a single paragraph giving out the topic. This is probably 15th paper I’m reviewing that attempts to establish anthropology - in vane. The term "identification" should be defined better.

We thank you for this remark; we have briefly defined the identification process.

How was the sex determined in in ln 116?

The collection is identified; as such, there is documentation regarding different biographical features of the individuals, including biological sex.

Some parts of the "introduction" belong to rhe "methods", and the methodology section should be sub-divided tnto "subheadings" (2??).

Subheadings were added, and methods were reviewed.

I believe that lns 86-88 comprise whole paper and my oppinion on it. If anything - this should be the "concllusion". Discussion is weak and mistakej for the "conclusion". Discussion should contain  the results and outcomes of a study. An effective discussion informs readers what can be learned from your experiment and provides context for the results.

We have deepened the discussion and enhanced the comparison with a previous study. We have also moved some text from the conclusion that fits better in the discussion.

"Calcified atherosclerotic plaques" are "CAP" in most cases, but somewhere (as in ln 89) it is given in full

Revised.

Ln 115 - "shortened sample"???

Revised.

Figures 1 and 2 are of a low quality and blurred - too poor for a publication and outlook/ color scheme should be changed. In figs. 3 and 4.

New images with better quality were added.

Round 2

Reviewer 3 Report

Comments and Suggestions for Authors

This version is significantly improved, both in the content and presentation. It is a shame that I don't see its real potential in forensic identification. However, using atherosclerotic artifacts in forensic anthropology might be a valuable tool in anthropology. 

Author Response

Thank you for your comments. We have developed more about the potential for identification of atherosclerotic plaques.